# Immobilization of Peroxo-Heteropoly Compound and Palladium on Hydroxyapatite for the Epoxidation of Propylene by Molecular Oxygen in Methanol

**DOI:** 10.3390/molecules28010024

**Published:** 2022-12-20

**Authors:** Yanyong Liu

**Affiliations:** Research Institute of Energy Process, National Institute of Advanced Industrial Science and Technology (AIST), AIST Tsukuba West, 16-1 Onogawa, Tsukuba 305-8569, Japan; yy.ryuu@aist.go.jp; Tel.: +81-(0)29-861-4826

**Keywords:** hydroxyapatite, peroxo-heteropoly compound, palladium, surface immobilization, propylene epoxidation, molecular oxygen, methanol

## Abstract

Peroxo-heteropoly compound PO_4_[W(O)(O_2_)_2_] was synthesized on calcium-deficient hydroxyapatite using a reaction of surface [HPO_4_]^2−^ groups on hydroxyapatite with a Na_2_[W_2_O_3_(O_2_)_4_] aqueous solution. The vibration of [HPO_4_]^2−^ at 875 cm^−1^ became very weak, and the vibration of the peroxo-oxygen bond [O–O]^2−^ at 845 cm^−1^ appeared in the FT-IR spectrum of the solid product, indicating that PO_4_[W(O)(O_2_)_2_] was formed on the surface of hydroxyapatite. The formed solid sample was further reacted with PdCl_2_(PhCN)_2_ in an acetone solution to fix PdCl_2_ between the O sites on the hydroxyapatite. Elemental analyses proved that the resultant solid contained 1.2 wt.% Pd, implying that PdCl_2_ molecules were immobilized on the surface of hydroxyapatite. The hydroxyapatite-based hybrid compound containing Pd and PO_4_[W(O)(O_2_)_2_] was used as a heterogeneous catalyst in a methanol solvent for propylene epoxidation by molecular oxygen in an autoclave batch reaction system. A propylene conversion of 53.4% and a selectivity for propylene oxide of 88.7% were obtained over the solid catalyst after reaction at 363 K for 8 h. The novel catalyst could be reused by a simple centrifugal separation, and the yield of propylene oxide did not decrease after the reaction for five runs. By prolonging the reaction time to 13 h, the highest yield of propylene oxide at 363 K over the solid catalyst was obtained as 53.8%, which was almost the same as that of the homogeneous catalyst containing PdCl_2_(PhCN)_2_ and [(C_6_H_13_)_4_N]_2_{HPO_4_[W(O)(O_2_)_2_]_2_} for the propylene epoxidation. Methanol was used as a solvent as well as a reducing agent in the propylene epoxidation by molecular oxygen. Small particles of Pd metal were formed on the surface of the hybrid solid catalyst during the reaction, and acted as active species to achieve the catalytic turnover of PO_4_[W(O)(O_2_)_2_] in the propylene epoxidation by molecular oxygen in methanol.

## 1. Introduction

Propylene oxide (denoted as PO), obtained from the epoxidation of propylene, is an important industrial intermediate to produce polyurethanes, unsaturated resins, and other products. The chlorohydrin process and hydroperoxide process are conventional industrial methods for producing PO from propylene. However, they are not environmentally benign methods due to the formation of a large amount of co-product CaCl_2_ or t-butyl alcohol. Hydrogen peroxide (H_2_O_2_) is an environmentally benign oxidant and exhibits a high selectivity for PO in propylene epoxidation [1]. However, H_2_O_2_ is too expensive to achieve an economically viable process. Molecular oxygen (O_2_) is the best oxidant from views of cost and environment. The catalytic oxidation of propylene by O_2_ mainly forms acrolein, and the selectivity for PO is very low due to the high activity of allylic C−H bonds of propylene molecules [2,3]. The selectivity for PO, in the propylene oxidation by O_2_, greatly increased in the presence of a reducing agent, such as H_2_ [4] and N_2_O [5]. In an industrial process, it is hard to handle the gaseous reducing agent H_2_ or N_2_O in the presence of O_2_. We developed a homogeneous catalytic system containing Pd and peroxo-heteropoly compound for the propylene epoxidation by O_2_ in methanol [6]. Methanol was used as a solvent as well as a reducing agent in the reaction. The catalytic system is promising in the industry because methanol is a tractable and reusable liquid-reducing agent [7]. Compared to homogeneous catalysts, heterogeneous catalysts are desirable in the industry because they are easily separated and reused [8,9,10]. We have designed two kinds of heterogeneous catalysts containing Pd and peroxo-heteropoly compound for the propylene epoxidation by O_2_ in methanol [11,12].

Heteropoly compounds are attractive catalysts for acid-catalytic and oxidation reactions in homogeneous and heterogeneous systems [13,14,15]. Moreover, the bifunctional catalysts, containing heteropoly compounds and noble metals, are effective catalysts for many industrially important reactions [16,17,18,19,20,21,22,23,24]. H_3_PW_12_O_40_ is an excellent catalyst for olefin epoxidation using H_2_O_2_ as an oxidant [15]. Peroxo-heteropoly compounds formed in situ from H_3_PW_12_O_40_ and H_2_O_2_ are the real active species in the catalytic systems [25]. It is important to effectively utilize peroxo-heteropoly compounds for catalytic oxidation reactions in industrial processes [26,27,28]. The immobilization of peroxo-heteropoly compounds, on solid supports by chemical bonds, has been used to develop heterogeneous catalysts [29,30,31,32,33]. The reaction of surface P−OH groups with peroxo-isopoly compound [W(O)(O_2_)_2_]_2_ to form peroxo-heteropoly compounds is an efficient method to immobilize peroxo-heteropoly compounds on the supports [11,29].

Hydroxyapatite (denoted as HAP) has been used as a kind of useful material in various fields [34]. HAP attracts considerable attention as an effective catalyst and support in organic synthesis [35]. HAP-supported heteropolyacids have been reported as excellent catalysts in the epoxidation of olefin using H_2_O_2_ as an oxidant [36,37,38]. Stoichiometric HAP has a formula of Ca_10_(PO_4_)_6_(OH)_2_, giving a Ca/P molar ratio of 1.67 [34]. Nonstoichiometric HAP is a kind of calcium-deficient HAP with a formula of Ca_10−x_(HPO_4_)_x_(PO_4_)_6−x_(OH)_2−x_, where x ranges from 0 to 2, giving a Ca/P molar ratio ranging from 1.67 to 1.33 [39,40,41,42]. The surface of calcium-deficient HAP contains the groups of HPO_4_, which can be used for surface modification [43]. On the other hand, PdCl_2_ can be immobilized on the HAP surface and the PdCl_2_-immobilized HAP has been used as a highly effective catalyst in alcohol oxidation by O_2_ [44,45,46,47,48].

This study reported a kind of novel hybrid compound prepared by immobilizing peroxo-heteropoly compound and PdCl_2_ on the HPA surface at the same time. The hybrid compound showed high catalytic ability and reusability in the epoxidation of propylene by O_2_ in methanol.

## 2. Results

### 2.1. FT-IR Spectra

Figure 1 shows the FT-IR spectra of various samples. HAP showed main peaks at around 1060, 875, 680, and 630 cm^−1^ in the FT-IR spectrum. These peaks coincided with the FT-IR spectrum of HAP in the literature [49]. The large peak at around 1060 cm^−1^ and its shoulder peaks were assigned to the vibrations of tetrahedral [PO_4_]^3–^ groups, and the peak at 875 cm^−1^ was assigned to the vibrations of [HPO_4_]^2−^ groups in the calcium-deficient hydroxyapatite HAP [50,51,52]. The FT-IR spectrum of K_2_[W_2_O_3_(O_2_)_4_] showed peaks at around 980, 960, 845, 775, and 615 cm^−1^, coinciding with the literature data [53]. The peak at 845 cm^−1^ was assigned to the stretching vibration of the peroxy-oxygen bond [O–O]^2−^ [54]. In the FT-IR spectrum of [(C_6_H_13_)_4_N]_2_{HPO_4_[W(O)(O_2_)_2_]_2_} (denoted as THA-PW_2_), the big peak at around 1070 cm^−1^, and its shoulder peaks, were assigned to the stretching vibration of P−O, and the peak at 930 cm^−1^ was assigned to the stretching vibration of W=O nearby P−O bonds [53]. Moreover, the peak at 845 cm^−1^ due to the stretching vibration of [O–O]^2−^ was observed in the FT-IR of THA-PW_2_. The sample of PO_4_[W(O)(O_2_)_2_]-immobilized HAP (denoted as PW-HAP) exhibited the main peaks of HAP and THA-PW_2_ in the FT-IR spectrum. However, the vibrations of the HPO_4_ units at 875 cm^−1^ almost disappeared, and the vibration of W=O nearby P−O bonds appeared at 930 cm^−1^ in the IR spectrum of PW-HAP. The stretching vibration of the peroxy-oxygen bond [O–O]^2−^ could be observed in the spectrum in the PW-HAP. The FT-IR spectrum of PdCl_2_ did not show an obvious peak in the range from 600 to 1400 cm^−1^. On the contrary, PdCl_2_(PhCN)_2_ showed many peaks correlating with the organic PhCN groups in the FT-IR spectrum. The IR spectrum of PdCl_2_-immobilized PW-HAP (denoted as Pd-PW-HAP) was almost the same as the IR spectrum of PW-HAP. The peaks of PhCN groups could not be observed in the IR spectrum of Pd-PW-HAP.

### 2.2. Elemental Analyses

Table 1 shows the results of elemental analyses of various HAP-containing samples. HAP contained 34.3 wt.% of Ca, and 19.8 wt.% of P, with a Ca/P molar ratio of 1.34. PW-HAP contained 26.8 wt.% of Ca, 15.6 wt.% of P, and 13.6 wt.% of W by element analyses. This indicated that W was successfully introduced in PW-HAP by stirring HAP powders in the Na_2_[W_2_O_3_(O_2_)_4_] aqueous solution. On the other hand, Pd-PW-HAP contained 26.1 wt.% of Ca, 15.2 wt.% of P, 13.2 wt.% of W, and 1.2 wt.% of Pd. This indicated that Pd was fixed on the surface of the Pd-PW-HAP sample. The molar ratio of Ca/P in each sample was almost the same as the designed Ca/P molar ratio (1.33) in the synthesis step. 

### 2.3. Epoxidation of Propylene by O_2_ in Methanol over Various Catalysts

Appendix A shows the GC charts of gas and liquid products after reaction at 363 K for 8 h over Pd-PW-HAP. 

Table 2 shows the results of propylene epoxidation by O_2_ in methanol over various catalysts. PW-HAP showed high selectivity for PO of 90.3%, but the C_3_H_6_ conversion was low (2.3%) after reaction at 363 K for 8 h. PW groups in PW-HAP contain peroxy-oxygen bonds [O–O]^2−^ which can stoichiometrically epoxidize olefin to olefin oxide [25]. However, the catalytic turnover of peroxy-oxygen bonds in PW-HAP could not be achieved by O_2_. PdCl_2_(PhCN)_2_ showed a very low C_3_H_6_ conversion of 2.1%, and a low selectivity for PO of 9.5% for the oxidation of propylene by O_2_. The solid catalyst Pd-PW-HAP showed a propylene conversion of 53.4% and a selectivity for PO of 88.7%, after reaction at 363 K for 8 h. The by-products were acrolein, propionaldehyde, acetone, propane, and other products (mainly ring-opened compounds of PO). The homogeneous catalyst Pd+PW_2_ (with 0.24 g [(C_6_H_13_)_4_N]_2_{HPO_4_[W(O)(O_2_)_2_]_2_} and 0.02 g PdCl_2_(PhCN)_2_) was designed to contain the same amounts of W and Pd as the heterogeneous Pd-PW-HAP catalyst. Pd+PW_2_ showed a propylene conversion of 57.2% and a selectivity for PO of 88.5% for the reaction. Pd-HMS-PW_2_ and PdMgAl-PW_4_ are two kinds of solid catalysts we reported previously for the propylene epoxidation by O_2_ in methanol [11,12]. As shown in Table 2, Pd-HMS-PW_2_ showed a propylene conversion of 44.6% and a selectivity for PO of 89.6%, and PdMgAl-PW_4_ showed a propylene conversion of 48.3% and a selectivity for PO of 87.1% after reaction at 363 K for 8 h. The turnover number (TON) of the catalyst was defined as the number of formed PO molecules per peroxy-oxygen bond [O–O]^2−^ in the catalyst. The number of the peroxy-oxygen bond [O–O]^2−^ was calculated using the catalyst’s W amounts, assuming one W molecule was bonding with two [O–O]^2−^ groups. The simultaneous existence of the peroxo-heteropoly compound and Pd is very important for improving the PO yield in the propylene epoxidation by O_2_ in methanol. As shown in Table 2, the PO yield and TON over the catalysts containing Pd and the peroxo-heteropoly compound decreased in an order of Pd+PW_2_ > Pd-PW-HAP > PdMgAl-PW_4_ > Pd-HMS-PW_2_ for the propylene epoxidation by O_2_ in methanol at 363 K. The homogeneous catalyst Pd+PW_2_ showed a higher PO yield and TON of [O–O]^2−^ than the heterogeneous catalysts. On the other hand, Pd-PW-HAP obtained the highest PO yield and TON of [O–O]^2−^ among various heterogeneous catalysts with the same loadings of Pd and W (1.2 wt.% of Pd and 13.2 wt.% of W).

### 2.4. Comparison of Catalytic Performance between Pd-PW-HAP and Pd+PW_2_

Figure 2 shows the effect of reaction time on the propylene epoxidation by O_2_ in methanol at 363 K over Pd-PW-HAP and Pd+PW_2_. The homogeneous catalyst Pd+PW_2_ exhibited a C_3_H_6_ conversion higher than that of the heterogeneous catalyst Pd-PW-HAP at the same reaction time. On the other hand, the PO selectivity over Pd+PW_2_ was higher than that over Pd-PW-HAP from 1 to 7 h, but became lower than that over Pd-PW-HAP when the reaction time was more than 8 h. The increase in by-products from the open-ring and over-oxidation decreased the PO selectivity in a long-time reaction. Pd+PW obtained the highest PO yield after the reaction for 10 h, and then the PO yield decreased with increasing reaction time. On the other hand, the highest PO yield over Pd-PW-HAP was obtained after the reaction for 13 h. The highest PO yield over Pd-PW-HAP (53.8%) was almost the same as that of Pd+PW_2_ for the propylene epoxidation at 363 K by O_2_ in methanol.

### 2.5. Stability of Solid Catalyst Pd-PW-HAP

Figure 3 shows the stability of the solid catalyst Pd-PW-HAP for the propylene epoxidation by O_2_ in methanol. The experiment was carried out using a method reported previously [11]. A portion of 0.5 g fresh Pd-PW-HAP catalyst was used in the first run for catalyzing the propylene epoxidation at 363 K for 8 h. After the reaction, the autoclave reactor was cooled down to room temperature. The used solid catalyst was obtained by centrifugal separation, and was then put into another reactor containing reactants and solvent. The second run was started by heating the reactor containing the used Pd-PW-HAP to 363 K. In the meantime, the liquid after the first run, by eliminating out Pd-PW-HAP, was charged with reaction gases for a further reaction at 363 K for 8 h. As shown in Figure 3, the PO yield over the used Pd-PW-HAP in the second run did not decrease in comparison with that over the fresh catalyst in the first run. On the contrary, the liquid without Pd-PW-HAP did not show a further reaction at 363 K, and the PO yield slightly decreased with reaction time due to the increase in ring-opened products. 

### 2.6. Reusability of Solid Catalyst Pd-PW-HAP

Figure 4 shows the time courses of solid catalyst Pd-PW-HAP in the propylene epoxidation by O_2_ in methanol for five runs. The used catalyst was obtained through centrifugal separation, and then dried under vacuum at room temperature for 3 h. When the amount of the used catalyst after drying was less than 0.5 g, due to loss in the operation, a small amount of fresh catalyst was added to the used catalyst to keep the catalyst amount at 0.5 g for the next run. The propylene conversion greatly increased, but the selectivity for PO slightly decreased with reaction time in each run. The catalytic activity of Pd-PW-HAP was almost kept in the reaction for five runs.

Table 3 shows the reusability of solid catalyst Pd-PW-HAP for the propylene epoxidation by O_2_ in methanol. The fresh catalyst showed a C_3_H_6_ conversion of 53.4% and a selectivity for PO of 88.7% after the reaction at 363 K for 8 h. The used catalysts showed very similar C_3_H_6_ conversion and selectivity for PO to the fresh catalyst after reaction at 363 K for 8 h. The PO yield obtained in the fifth run was 47.3%, which was the same as those obtained in the first run.

### 2.7. XRD Patterns of Various Samples

Appendix A shows XRD patterns of HAP and Pd-PW-HAP before reaction. In the range from 5 to 70 degrees, the samples showed characteristic reflections of the HAP phase according to the Rigaku PDXL2 database (No 1011242). The strong reflections of HAP and Pd-PW-HAP existed in the range of 25–45 degrees in the XRD patterns [55,56]. 

Figure 5 shows the XRD patterns of Pd-PW-HAP and Pd+PW_2_ before and after reaction. The sample of Pd+PW_2_ before reaction showed the reflections of the peroxo-heteropoly compound in the XRD pattern. For the sample of Pd+PW_2_ after reaction at 363 K for 8 h, a weak peak at 40.1 degrees corresponding to the reflection of (1 1 1) for Pd^0^ metal species could be observed in the XRD pattern. The XRD pattern of Pd-PW-HAP, after reaction at 363 K for 8 h, was almost the same as that of Pd-PW-HAP before reaction. Because a peak at 39.9 degrees corresponding to the (3 1 0) reflection of HAP appeared in the XRD pattern, the reflection of Pd (1 1 1) at 40.1 degrees could not be ensured in the XRD pattern of Pd-PW-HAP after reaction, due to overlapping by the (3 1 0) reflection of HAP. 

### 2.8. TEM Images of Pd-PW-HAP before and after Reaction

Figure 6 shows the TEM images of Pd-PW-HAP before reaction and after reaction. For the sample of Pd-PW-HAP before reaction, the metal particles could not be observed in the TEM image. On the hand, for the sample of Pd-PW-HAP after reaction at 363 K for 8 h, the metal particles were observed as small black spots in the TEM image. The diameter of metal particles ranged from 1 to 2 nm in the TEM image of Pd-PW-HAP after reaction, and the size distribution of metal particles was uniform. Moreover, the particle size of Pd metal did not increase after reaction at 363 K for 8 h for five runs.

Appendix A shows the results of elemental analyses of Pd-PW-HAP before reaction by ICP and EDS. Three points on the Pd-PW-HAP particle in the TEM image of Pd-PW-HAP before reaction (Figure 6A) were analyzed using the EDS instrument attached to the electron microscope. The three points by EDS analyses gave a similar composition of each element, indicating that PW and Pd were uniformly distributed on the surface of Pd-PW-HAP before reaction. Because EDS mainly analyzed the solid surface of Pd-PW-HAP, but ICP analyzed the HNO_3_ aqueous solution of Pd-PW-HAP, the EDS analyses gave lower values of Ca and P and higher values of W and Pd in comparison with the results of ICP analyses. 

### 2.9. EXAFS Functions of Pd-PW-HAP before and after Reaction

Figure 7 shows the EXAFS functions of various samples. The sample of Pd-PW-HAP before reaction showed a Pd–Cl interaction peak at 1.85 Å, coinciding with the Pd–Cl interaction peak in the PdCl_2_ sample. As for the sample of Pd-PW-HAP after reaction at 363 K for 8 h, the Pd–Cl interaction peak disappeared and an interaction peak at 2.3 Å was observed in the EXAFS function. In comparison with the EXAFS function of Pd foil, the interaction peak at 2.3 Å was assigned to the interaction of the Pd–Pd bond in the Pd metal. The PdCl_2_ molecules immobilized on the Pd-PW-HAP were reduced to the Pd^0^ metal species during the reaction.

### 2.10. XPS Spectra of Pd-PW-HAP before and after Reaction

Figure 8 shows the XPS spectra of Pd-PW-HAP before and after reaction. The binding energy of the Pd(3d_5/2_) peak was 338.2 eV in the XPS spectrum of Pd-PW-HAP before reaction, indicating that the Pd species remained in the oxidation state of +2 in the sample. On the other hand, the sample of Pd-PW-HAP after reaction showed the binding energy of the metallic Pd^0^ species with 3d_5/2_ of 334.9 eV and 3d_3/2_ of 340.2 eV in the XPS spectrum. These results proved that all Pd^2+^ species in the sample of Pd-PW-HAP before reaction were reduced into the metallic Pd^0^ species during the epoxidation of propylene by O_2_ in methanol at 363 K for 8 h.

### 2.11. Solvent Effect in the Epoxidation of Propylene by O_2_ over Pd-PW-HAP

Table 4 shows the results of catalytic propylene epoxidation by O_2_ over Pd-PW-HAP in various solvents. Pd-PW-HAP obtained a propylene conversion of 53.4%, and a selectivity for PO of 88.7% after reaction at 363 K for 8 h in the methanol solvent. On the other hand, in the chloroform solvent, Pd-PW-HAP showed a low propylene conversion of 2.2% although the selectivity for PO was high (91.5%) after reaction at 363 K for 8 h. Hence, the peroxo-heteropoly compound in Pd-PW-HAP stoichiometrically epoxidizes propylene to PO using the peroxy-oxygen bonds, but the catalyst turnover could not be achieved in the chloroform solvent. Methanol is a necessary solvent for the epoxidation of propylene by O_2_ over Pd-PW-HAP. 

### 2.12. Co-Products from Methanol in the Epoxidation of Propylene by O_2_

Table 5 shows the consumed amounts, and formed amounts, in the propylene epoxidation in methanol over various catalysts at 363 K for 8 h. PW-HAP consumed a small number of C_3_H_6_ (0.7 mmol) and selectively converted them to PO (0.6 mmol). The methanol medium was not consumed over PW-HAP during the reaction. These results indicated that PW-HAP stoichiometrically converted C_3_H_6_ to PO using the peroxy-oxygen bonds in PW, but the peroxy-oxygen bonds were not recovered to achieve a catalytic turnover. On the other hand, Pd-PW-HAP consumed 18.8 mmol of C_3_H_6_ to form 17.0 mmol of PO after reaction at 363 K for 8 h. Hence, Pd is a necessary component in the catalysts containing peroxo-heteropoly compounds for the epoxidation of propylene by O_2_ in methanol. As shown in Table 5, an amount of 10.1 mmol of methanol was consumed to form co-products (mainly CO and CO_2_) over Pd-PW-HAP after reaction at 363 K for 8 h. O_2_ was consumed for both oxidizing propylene and oxidizing methanol during the reaction. The consumed methanol was almost converted to the co-products involving CO, CO_2_, and small amounts of HCHO and HCH(OH)_2_. 

## 3. Discussion

### 3.1. Synthesis Route of Pd-PW-HAP

Figure 9 shows the route in the synthesis of the hybrid compound Pd-PW-HAP. 

The first step in the synthesis route is grafting peroxotungstic ions on the HAP surface to form PW-HAP by using the reaction of surface HPO_4_^2−^ groups. Calcium-deficient HAP has a structure of stoichiometric HAP (Ca/P = 1.67), but involves Ca and OH defects and HPO_4_^2−^ groups [44,45,46,47]. The surface HPO_4_^2−^ groups have activity for some chemical reactions, and thus have been used for surface modification to graft other groups [38]. On the other hand, the HPO_4_^2−^ groups on the solid surface have a reaction activity with peroxotungstic ions to form peroxo-heteropoly ions in an aqueous solution [11,27,28]. As shown in the FT-IR spectra (Figure 1), HAP showed the vibrations of HPO_4_ groups at 875 cm^−1^, which proved the existence of the HPO_4_ group in the calcium-deficient HAP. Moreover, the vibrations at 875 cm^−1^ became weak and the vibrations of the peroxo-bond O–O at 845 cm^−1^ appeared in the FT-IR spectrum of PW-HAP. This indicated the number of HPO_4_ groups on HAP greatly decreased by reacting with [W_2_O_3_(O_2_)_4_]. In comparison with the IR spectra of PW-HAP and K_2_[W_2_O_3_(O_2_)_4_], the vibration of W=O nearby P−O bonds at 940 cm^−1^ appeared in the IR spectra of PW_2_-HAP. This implied that the HPO_4_ groups reacted with [W_2_O_3_(O_2_)_4_] ions to form peroxo-heteropoly compounds on the PW-HAP. 

The second step in the synthesis route is fixing PdCl_2_ molecules between the O atoms of PO_4_^3–^ groups on the HAP surface. It has been reported that PdCl_2_ molecules can be fixed on the HAP surface, and the resultant compounds are excellent catalysts for alcohol oxidation by O_2_ [44,47,48]. As shown in the FT-IR spectra (Figure 1), the peaks of PhCN groups disappeared in the FT-IR spectrum of Pd-PW-HAP, indicating that the organic PhCN groups dropped out from Pd-PW-HAP during the synthesis process. Because the IR spectrum of Pd-PW-HAP was almost the same as that of PW-HAP, the peroxo-heteropoly compound PW remained on the surface of HAP during the PdCl_2_ immobilization. Using the two steps of surface modification, both peroxo-heteropoly compounds and PdCl_2_ molecules were immobilized on the HAP surface in Pd-PW-HAP simultaneously.

The molecular formulas of the samples in Table 1 can be obtained using the elemental compositions and the general formulas. At first, the total molecular weight of each compound was calculated using the content of Ca from elemental analysis, and the chemical stoichiometry of Ca in the general formula. Then, the chemical stoichiometries of various groups were calculated using their proportions in the sample, total molecular weight, and the general formula. The deficiency between the summed percentages of various elements and 100% was calculated as adsorbed water. By calculation using the method described above, the formulas of HPA, PW-HAP, and Pd-PW-HAP were Ca_4_(HPO_4_)(PO_4_)_2_·1.1H_2_O, Ca_4_(HPO_4_)_0.56_{HPO_4_[W(O)(O_2_)]_0.44_}(PO_4_)_2_·1.9H_2_O, and Ca_4_(HPO_4_)_0.56_{HPO_4_[W(O)(O_2_)_2_}_0.44_(PO_4_)_2_(PdCl_2_)_0.07_·2.1H_2_O, respectively. It can be supposed that the HPO_4_ groups on the HAP surface almost converted to the PW groups in PW-HAP because a large excessive amount of Na_2_[W_2_O_3_(O_2_)_4_] was used in the synthesis step by reacting HAP with Na_2_[W_2_O_3_(O_2_)_4_]. However, the HPO_4_ groups in the inner structure of HAP remained in the PW-HAP. On the other hand, the actual Pd amount in Pd-PW-HAP (as shown in Table 1) was 1.2 wt.%, which was lower than the designed Pd amount in Pd-PW-HAP (2 wt.%) in the synthesis step. This implied that a suitable distance and angle between two PO_4_ groups were needed to fix PdCl_2_ molecules on the HAP surface. 

### 3.2. Strong Points of Pd-PW-HAP

As shown in the XRD pattern Pd-PW-HAP before reaction (in Figure 5), the reflections of palladium and peroxo-heteropoly compounds did not appear. The crystals of PW and PdCl_2_ did not form on the Pd-PW-HAP, implying that PW and PdCl_2_ were uniformly immobilized on the surface of HAP. Because palladium and peroxo-heteropoly compounds were fixed on the surface of HAP by chemical bonds, the active components (PW and Pd) did not leach from Pd-PW-HAP in the solvent during the reaction (results in Figure 3). Not only the Pd^2+^ complex could be fixed on the surface of HAP [44], but also the Pd^0^ nanocluster formed from the reduction of Pd^2+^ during the reaction had a strong interaction with the surface of HAP [48]. This is the reason why Pd-PW-HAP could be used for five runs without a decrease in the catalytic activity (as shown in Table 3). Because the homogeneous catalyst Pd+PW_2_ easily diffuses in the methanol solvent, Pd+PW_2_ showed a higher C_3_H_6_ conversion than that of Pd-PW-HAP (as shown in Table 2). However, the only possible method for recovering the homogeneous catalysts was vacuum distillation of the mixture after the reaction. The vacuum distillation consumed a larger amount of energy compared with the centrifugal separation. On the contrary, the solid catalyst Pd-PW-HAP can be reused by simple centrifugal separation in the propylene epoxidation by O_2_ in methanol. Hence, Pd-PW-HAP is desirable in industry manufacture because of its ease of separation and renewability. As shown in Figure 2, there was only a small difference in the PO yield between Pd-PW-HAP and Pd+PW_2_ because PW and PdCl_2_ were uniformly distributed on the HAP surface. Moreover, the highest PO yield over Pd-PW-HAP, obtained by increasing reaction time, was equal to the highest PO yield over Pd+PW_2_ for the reaction at 363 K (as shown in Figure 2b). HAP-immobilized Pd is a kind of highly efficient catalyst for oxidation reactions using O_2_ as an oxidant [44,47,48]. This is the reason that Pd-PW-HAP showed a catalytic activity higher than those of PdMgAl-PW_4_ and Pd-HMS-PW_2_ for the propylene epoxidation by O_2_ in methanol (as shown in Table 2).

### 3.3. Active Pd Species in Pd-PW-HAP

As shown in the XRD pattern of Pd+PW_2_ after reaction (Figure 5), the peak at 40.1 degrees, corresponding to Pd^0^ species, could be observed. The metal Pd^0^ species was the active species in the propylene epoxidation by O_2_ in methanol over the homogenous catalyst Pd+PW_2_ [7]. On the other hand, the peak of Pd^0^ species at 40.1 degrees could not be confirmed in the XRD pattern of Pd-PW-HAP after reaction (as shown in Figure 4). It is necessary to confirm the active Pd species in the propylene epoxidation by O_2_ in methanol over the heterogeneous catalyst Pd-PW-HAP. The EXAFS function confirmed that the PdCl_2_ species were converted to Pd^0^ metal species after the reaction (as shown in Figure 7). Furthermore, the Pd metal particles were observed in the TEM image of Pd-PW-HAP after the reaction (as shown in Figure 6). The size of Pd^0^ metal particles ranged from 1–2 nm in the TEM image of Pd-PW-HAP after the reaction. The metal particle with a size lower than 3 nm hardly exhibited their reflections in the XRD pattern [57]. Hence, the reflections of Pd^0^ metal did not appear in the XRD pattern of Pd-PW-HAP after reaction because the size of the formed Pd^0^ metal particles was too small. The small Pd^0^ particles were formed after reaction because the precursor PdCl_2_ molecules were uniformly fixed on the surface of Pd-PW-HAP before reaction. As a result, Pd^0^ metal was the species for the propylene epoxidation by O_2_ in methanol over the solid catalyst Pd-PW-HAP. 

### 3.4. Roles of Methanol in the Reaction

Figure 1 shows the co-oxidation of methanol in the system containing Pd and O_2_. HAP-immobilized palladium is a kind of highly efficient catalyst for the oxidation of alcohols by O_2_ [44,45,46,47,48]. Methanol has the highest reducibility among various alcohols, and thus can be oxidized to a peroxy intermediate HOCH_2_OOH by O_2_ in the presence of a Pd catalyst [58,59]. The aliphatic organic peroxide HOCH_2_OOH can achieve the catalytic turnover of the peroxo-heteropoly compound in propylene epoxidation [15,25]. Some HOCH_2_OOH molecules turn back to HCH(OH)_2_ for the next catalytic cycle after giving their peroxo-bonds to heteropoly compounds. On the other hand, some HOCH_2_OOH molecules decompose to CO_x_ and H_2_O in the catalytic system because HOCH_2_OOH is not stable [60]. Hence, methanol was not only a solvent but also a co-reactant or reducing agent in the propylene epoxidation by O_2_ in methanol, over Pd-PW-HAP. Because the reaction temperature was low (mainly at 363 K) in this study, only a small amount of methanol underwent co-oxidation by O_2_ during the reaction.

### 3.5. Reaction Mechanism for the Reaction

Figure 2 shows the reaction mechanism for propylene epoxidation by O_2_ in methanol. The peroxo-heteropoly compounds in PW-HAP stoichiometrically epoxidize propylene to PO by O–O peroxo-bonds in methanol, but the O–O peroxo-bonds cannot be recovered by O_2_ (results in Table 2). Moreover, the O–O peroxo-bonds in the peroxo-heteropoly compounds on Pd-PW-HAP cannot be recovered in a chloroform solvent (results in Table 4). Hence, Pd on HAP oxidizes CH_3_OH to the peroxy intermediate HOCH_2_OOH by O_2_, and the formed HOCH_2_OOH acts to recover the O–O peroxo-bonds of peroxo-heteropoly compounds. Pd and methanol convert the weak oxidant O_2_ to the strong oxidant HOCH_2_OOH in situ during the reaction. As shown in Table 2, PdCl_2_(PhCN)_2_ showed a very low PO yield (0.2%) for the propylene epoxidation in methanol. This indicates that the concentration of HOCH_2_OOH formed in the system was low, and could not directly result in epoxidation of propylene to PO. The peroxo-heteropoly compounds are necessary for the catalytic system to convert propylene to PO using their peroxy-oxygen bonds. As a result, peroxo-heteropoly compounds achieve the catalytic conversion of propylene to PO, and Pd achieves the catalytic conversion of methanol and O_2_ to peroxy intermediate HOCH_2_OOH, which are used for recovering peroxo-heteropoly compounds in situ.

Although the aliphatic organic peroxide HOCH_2_OOH formed in situ was used as an oxidant for propylene epoxidation, the catalytic system in propylene epoxidation by O_2_ in methanol, over Pd-PW-HAP, is different from the conventional hydroperoxide process. In comparison with the t-butyl alcohol co-product formed in the conventional hydroperoxide process, the co-product CO_x_ can be used to synthesize methanol in the industrial process over Cu-based catalysts, which achieves the recycling of methanol solvent in the catalytic system. Pd-PW-HAP is a kind of heterogeneous catalyst that can be reused by simple centrifugal separation after the reaction. Because Pd-PW-HAP fixed Pd and PW on the HAP surface by chemical bonds, Pd-PW-HAP did not leach the active components to the solvent during the reaction. 

## 4. Materials and Methods

### 4.1. Reagents 

Inorganic reagents were purchased from Wako Pure Chemical Industries Ltd. (Tokyo, Japan) with purities higher than 99%. Organic reagents were purchased from Tokyo Chemical Industry Co., Ltd. (Tokyo, Japan) with purities higher than 99.5%. Gas cylinders were purchased from Sumitomo Seika Chemicals Co., Ltd. (Tokyo, Japan) with purities higher than 99.995%. 

### 4.2. Catalyst Preparation 

According to the literature, calcium-deficient HAP was prepared using a wet method [41]. A 1.0 M Ca(NO_3_)_2_ aqueous solution and a 0.6 M (NH_4_)_2_HPO_4_ aqueous solution were simultaneously added to 500 mL distilled water in a three-necked flask while stirring. The pH was kept at around 11 during the process by adding an ammonia solution. The resultant milky solution was further stirred at 353 K for 1 h. Then, the precipitate was filtered and washed with distilled water. The resulting solid was dried at 383 K for 24 h and calcined at 773 K for 3 h. The designed Ca/P molar ratio was 1.33.

Na_2_[W_2_O_3_(O_2_)_4_] aqueous solution was prepared by dissolving solid Na_2_WO_4_ in 30% H_2_O_2_ [53]. A portion of 15 g Na_2_WO_4_ powder was gradually added to 60 mL of 30% H_2_O_2_ at 323 K while stirring to form a clear solution. The solution was stored in a refrigerator at 278 K. 

K_2_[W_2_O_3_(O_2_)_4_] was obtained by adding an excessive amount of KCl to an aqueous solution of Na_2_[W_2_O_3_(O_2_)_4_] to form a K_2_[W_2_O_3_(O_2_)_4_] precipitate [54]. The precipitate was filtered, dried in the air at room temperature, and stored in a refrigerator at 278 K.

PW-HAP was prepared using a surface reaction, according to the method in the literature [11,29]. Two grams of HAP powder were added to 20 mL of Na_2_[W_2_O_3_(O_2_)_4_] solution with vigorous stirring at room temperature for 6 h. Then, the solid was isolated by filtration, dried in the air at room temperature, and stored in a refrigerator at 278 K. 

Pd-PW-HAP was synthesized by fixing PdCl_2_ molecules on PW-HAP, according to the method in the literature [44]. Two grams of PW-HAP were stirred in a 150 mL acetone solution of PdCl_2_(PhCN)_2_ (2.5×10^−3^ M) at room temperature for 3 h. The solid product Pd-PW-HAP was isolated by filtration, dried in the air at room temperature, and stored in a refrigerator at 278 K. The designed Pd loading was 2 wt.%.

THA-PW_2_, with a molecular formula of [(C_6_H_13_)_4_N]_2_{HPO_4_[W(O)(O_2_)_2_]_2_}, was prepared according to the method in the literature [54]. A portion of 2.5 g H_2_WO_4_ was added to 7 mL 30% H_2_O_2_. After stirring at 323 K for 40 min, 0.85 mL of 6 M H_3_PO_4_ was added to the supernatant. After further stirring for 5 min, 7.8 g of tetrahexylammonium chloride dissolved in 10 mL of water was added to the clear solution. Further stirring was carried out at room temperature for 10 min. Then, the solid product was filtered, dried in the air at room temperature, and stored in a refrigerator at 278 K. 

Pd-HMS-PW_2_, a mesoporous silica containing peroxo-heteropoly compound and Pd, was prepared according to the literature [11]. The peroxo-heteropoly compound {HPO_4_[W(O)(O_2_)_2_]_2_} was synthesized on the surface of HMS by reacting HMS-PrNH(PO_3_H_2_) with H_2_[W_2_O_3_(O_2_)_4_] aqueous solution, and then palladium ions were exchanged into the channels of mesoporous silica HMS. 

PdMgAl-PW_4_ was prepared by treating PdMgAl-[PW_11_O_39_] with 30% H_2_O_2_ according to the literature [12]. PdMgAl-[PW_11_O_39_] was synthesized using an ion exchange method to introduce heteropoly anion [PW_11_O_39_]^7−^ in the interlayer zone of Pd-containing MgAl-type hydrotalcite. The molar ratio of Mg/Al was three in the hydrotalcite.

### 4.3. Catalyst Characterization 

The elemental analyses were measured by a method of inductively coupled plasma (ICP) using a Thermo Jarrel Ash IRIS/AP instrument. The samples for the ICP analyses were obtained by dissolving HAP-based samples in the HNO_3_ aqueous solution.

The X-ray diffraction (XRD) patterns were measured by a RINT-2500HLR diffractometer (Rigaku Corp.) with Cu Kα radiation (λ = 0.154 nm). The operation voltage was 40 kV, and the operation current was 50 mA in the XRD analyses. 

Fourier transform infrared (FT-IR) spectra were carried out using a KBr pellet method by a JASCO FT/IR-7000 spectrometer. The KBr pellet, containing 0.5 wt.% of each sample, was measured at room temperature under atmospheric conditions. 

Transmission electron microscopy (TEM) images were taken using a JEOL JEM 2010FX electron microscope equipped with a Hitachi/Keves H-8100/Delta IV EDS at 200 kV. The samples were supported on carbon-coated copper grids before the TEM measurement.

X-ray absorption fine structure (XAFS) of Pd K-edges was measured at the beamline 10B of the Photon Factory in the National Laboratory for High-Energy Physics, Tsukuba, Japan. Extended X-ray Absorption Fine Structure (EXAFS) data were examined using the Rigaku EXAFS analysis program. Fourier transformation of k^3^-weighted normalized EXAFS data was carried out in the range of 3.5 Å < k/Å^−1^ < 11 Å.

X-ray photoelectron spectra (XPS) were carried out using a Phi-5500 ESCA spectrometer equipped with Mg Kα radiation (1253.6 eV). The chamber pressure was 10^−9^ Torr. The binding energy of Pd was calibrated by the C_1s_ of 284.6 eV.

### 4.4. Reaction Procedure and Analyses 

The reaction was performed in a 100 mL stainless steel autoclave reactor. In a typical reaction, 0.5 g Pd-PW-Hap and 20 mL methanol were added to the autoclave. A nitrogen gas with a pressure of 0.4 MPa was charged in the autoclave, and then charged out from the autoclave to change the air. After purging with nitrogen five times, the autoclave contained 0.1 MPa of nitrogen at room temperature. Then, 1.0 MPa of propylene and 1.0 MPa of oxygen were sequentially charged in the autoclave at room temperature. Then, the autoclave was immersed in an oil bath with temperature control. The reaction started with vigorous stirring (500 revolutions per minute) after the autoclave reactor was heated to the reaction temperature.

After the reaction, the autoclave was cooled down to room temperature. The gas in the autoclave was collected in a plastic gas bag. Then, a certain amount of 1,4-dioxane was added to the autoclave as an internal standard [61]. Inorganic gases (CO_2_, CO, O_2_, and N_2_) were analyzed by a Shimadzu 2014 TCD-GC equipped with a Shincarbon-ST packed column in He carrier gas. Organic gaseous compounds were analyzed by a Shimadzu 2014 FID-GC equipped with an RT-QPLOT capillary column. The factors of various gaseous compounds were obtained using a standard mixed gas (with a known concentration for each component) from a cylinder. On the other hand, the liquid and the solid were separated using a centrifuge. The liquid compounds were analyzed by an Agilent 6890 N FID-GC equipped with a PoraPLOT U capillary column.

The propylene conversion was calculated from the difference in its molar amounts before and after reactions using Equation (1):X_C3H6_ = (M_C3H6, before_ − M_C3H6, after_)/M_C3H6, before_ × 100(1)

The selectivity for each product was expressed in terms of carbon efficiency and was calculated using Equation (2):S = *n_i_* C_i_/Σ(*n_i_* C_i_) × 100(2)

Herein, *n_i_* denotes the number of carbon atoms in each product.

## 5. Conclusions

The novel hybrid compound Pd-PW-HAP was synthesized by immobilizing PdCl_2_ molecules and peroxo-heteropoly compounds on the surface of calcium-deficient hydroxyapatite. Pd-PW-HAP was an effective heterogeneous catalyst for propylene epoxidation by O_2_ in methanol. The propylene conversion over Pd-PW-HAP was slightly lower than that over the homogeneous catalyst in the epoxidation of propylene by O_2_ in methanol. By increasing the reaction time, Pd-PW-HAP could obtain the highest PO yield, comparable to that of the homogeneous catalyst at 363 K. Pd-PW-HAP could be used five times, without a decrease in the catalytic performance in the epoxidation of propylene. During the reaction, the immobilized PdCl_2_ formed small particles of Pd^0^ metal species on the Pd-PW-HAP surface by the reduction of methanol. During the reaction, a small part of the methanol solvent was oxidized to the peroxy intermediates HOCH_2_OOH by O_2_ in the presence of Pd. The peroxy intermediates recovered the peroxy-oxygen bonds, and achieved the catalytic turnover of peroxo-heteropoly compounds for the epoxidation of propylene.

## Data Availability

Not applicable.

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
