# Peer review of "Immobilization of Peroxo-Heteropoly Compound and Palladium on Hydroxyapatite for the Epoxidation of Propylene by Molecular Oxygen in Methanol"

_molecules, 2022, doi:10.3390/molecules28010024_

Round 1

Reviewer 1 Report

The article was very well written and science is of high standard. 

Not necessary, but it would have been good to do an XPS of the Pd-PW-HAP before and after catalysis, it would also confirm the reduction of the Pd and if any Pd(II) remained would also be detected (the ratio could be determine).

Author Response

 I added a new Figure (Figure 8. XPS spectra of Pd-PW-HAP before and after reaction) in the revised manuscript.

 I added the following part from Line 266 on Page 8 in the revised manuscript.

2.10. XPS spectra of Pd-PW-HAP before and after reaction

    Figure 8 shows the XPS spectra of Pd-PW-HAP before and after reaction. The binding energy of the Pd(3d5/2) peak was 338.2 eV in the XPS spectrum of Pd-PW-HAP before reaction, indicating that the Pd species remained in the oxidation state of +2 in the sample. On the other hand, the sample of Pd-PW-HAP after reaction showed the binding energy of the metallic Pd0 species with 3d5/2 of 334.9 eV and 3d3/2 of 340.2 eV in the XPS spectrum. These results proved that all Pd2+ species in the sample of Pd-PW-HAP before reaction were reduced into the metallic Pd0 species during the epoxidation of propylene by O2 in methanol at 363 K for 8 h.

 I added the following part to introduce the experiment of XPS

From Line 514 on Page 14 in the revised manuscript.

    X-ray photoelectron spectra (XPS) were carried out using a Phi-5500 ESCA spectrometer equipped with Mg Kα radiation (1253.6 eV). The chamber pressure was 109 Torr. The binding energy of Pd was calibrated by the C1s of 284.6 eV.

Reviewer 2 Report

Recommendation: Publish in Molecules after minor revision

The present paper is devoted to the synthesis of a hybrid catalyst for the epoxidation of propylene by molecular oxygen in methanol, which certainly attracts the interest of chemists and technologists.

In the current work the author extends this approach to supported heterogeneous catalysts based on immobilized peroxo-heteropoly compound and palladium on hydroxyapatite. Such hybrid system (Pd-PW-HAP) was an effective heterogeneous catalyst. The author provides a thorough analysis of the catalyst, the exact composition of the products is established, and a mechanism is proposed.

a)       In order to help the Author to improve their manuscript, I would like to draw his attention to some minor aspects. Previously, the author published a number of papers on the oxidation of propylene with oxygen in methanol:

1.       Journal of Catalysis 220 (2003) 513–518

2.       Catalysis Letters Vol. 93, Nos. 1–2, March 2004

3.       Applied Catalysis B: Environmental 58 (2005) 51–59

4.       Journal of Catalysis 248 (2007) 277–287

I would recommend comparing not only homogeneous catalysts, but also heterogeneous. It is necessary to show the advantages of the new catalyst.

b)      In Table 3, in addition to propylene conversion and propylene oxide selectivity, the yield of propylene oxide can be reported.

Author Response

a). As for the comparison of heterogeneous catalysts for propylene epoxidation.

In the revised manuscript, I added the reaction results of other solid catalysts containing Pd and the peroxo-heteropoly compound (PdMgAl-PW4 and Pd-HMS-PW2) in Table 2 and compared the reaction results of them with those of Pd-PW-HAP.

I added the following sentences in the revised manuscript.

From Line 145 on Page 4 in the revised manuscript.

As shown in Table 2, the PO yield and TON over the catalysts containing Pd and the peroxo-heteropoly compound decreased in an order of Pd+PW2 > Pd-PW-HAP > PdMgAl-PW4 > Pd-HMS-PW2 for the propylene epoxidation by O2 in methanol at 363 K. The homogeneous catalyst Pd+PW2 showed a higher PO yield and TON of [O–O]2– than the heterogeneous catalysts. On the other hand, Pd-PW-HAP obtained the highest PO yield and TON of [O–O]2– among various heterogeneous catalysts with the same loadings of Pd and W (1.2 wt.% of Pd and 13.2 wt.% of W).

    I added the following sentences to discuss the catalytic activity of Pd-PW-HAP among various heterogeneous catalysts.

From Line 376 on Page 11 in the revised manuscript.

As shown in Figure 2, there was only a small difference in the PO yield between Pd-PW-HAP and Pd+PW2 because PW and PdCl2 were uniformly distributed on the HAP surface. Moreover, the highest PO yield over Pd-PW-HAP obtained by increasing reaction time was equal to the highest PO yield over Pd+PW2 for the reaction at 363 K (as shown in Figure 2b). HAP-immobilized Pd is a kind of highly efficient catalyst for oxidation reactions using O2 as an oxidant [44,47,48]. This is the reason that Pd-PW-HAP showed a catalytic activity higher than those of PdMgAl-PW4 and Pd-HMS-PW2 for the propylene epoxidation by O2 in methanol (as shown in Table 2).

  1. Mori, K.; Hara, T.; Oshiba, M.; Mizugaki, T.; Ebitani, K; Kaneda, K. Catalytic investigations of carbon-carbon bond-forming reactions by a hydroxyapatite-bound palladium complex. New J. Chem. 2005, 29, 1174–1181.
  2. Mori, K.; Yamaguchi, K.; Hara, T.; Mizugaki, T.; Ebitani, K.; Kaneda, K. Controlled synthesis of hydroxyapatite-supported palladium complexes as highly efficient heterogeneous catalyst. J. Am. Chem. Soc. 2002, 124, 11572–11573.
  3. Hara, T.; Mori, K.; Oshiba, M.; Mizugaki, T.; Ebitani, K.; Kaneda, K. Highly efficient dehalogenation using hydroxyapatite-supported palladium nanocluster catalyst with molecular hydrogen. Green Chem. 2004, 6, 507–509.

I added the following part to introduce the syntheses of PdMgAl-PW4 and HMS-PW2

From Line 485 on Page 14 in the revised manuscript.

Pd-HMS-PW2, a mesoporous silica containing peroxo-heteropoly compound and Pd, was prepared according to the literature [11]. The peroxo-heteropoly compound {HPO4[W(O)(O2)2]2} was synthesized on the surface of HMS by reacting HMS-PrNH(PO3H2) with H2[W2O3(O2)4] aqueous solution, and then palladium ions were exchanged into the channels of mesoporous silica HMS.

PdMgAl-PW4 was prepared by treating PdMgAl-[PW11O39] with 30% H2O2 according to the literature [12]. PdMgAl-[PW11O39] was synthesized using an ion exchange method to introduce heteropoly anion [PW11O39] 7‒ in the interlayer zone of Pd-containing MgAl-type hydrotalcite. The molar ratio of Mg/Al was 3 in the hydrotalcite.

  1. b)   As for the yield of propylene oxide in Table 3

    I added the data on PO yield in Table 3 in the revised manuscript.

I added the following sentences from Line 206 on Page 6 in the revised manuscript.

The used catalysts showed very similar C3H6 conversion and selectivity for PO to the fresh catalyst after reaction at 363 K for 8 h. The PO yield obtained in the fifth run was 47.3%, which was the same as those obtained in the first run.

Reviewer 3 Report

The work was focused on the investigation of “Immobilization of peroxo-heteropoly compound and palla-dium on hydroxyapatite for the epoxidation of propylene by molecular oxygen in methanol”that may interesting to some readers for some meaningful results. It can be recommended to publish on the journal of molecules after revised.

Comments:

1.   The range of XRD patterns are too narrow, normally, XRD patterns are range from 5 to 90°for inorganic materials. And the peaks should be indentified to the reference JCPDS. Figure4 show a peak at 40°of A and B that are the same with D, why it is not belong to Pd?

2.   The EDS mapping of TEM characterization is needed for the verification of Pd distribution. The TEM image of Pd-PW-HAP after fifth circle reaction is also needed, for the certification of the stability of Pd-PW-HAP catalyst.  

3.   Why Pd particles can be fix on the PW-HAP after reaction at 363K for several circles. Weather Pa was coordinated with ions of PW-HAP, and is it exposed on the surface? Weather the characterization of XPS is needed.

4.   Exafs spectra should be showed the figure 6.

5.   The primary data of chromatography should be provided for review.

6.   The TOF of Pd is very important for the catalyst performance that should be calculated and added in the revised version.

Author Response

1. In the revised manuscript, I added a new Figure with degrees from 5 to 70 in the XRD patterns (Figure S2. XRD patterns of HAP and Pd-PW-HAP before reaction), and indicated the miller indices of main reflections according to the reference database.

    Many reflections appeared at degrees above 55° in the XRD pattern of HAP-based compounds but all of them were weak. In the XRD patterns of HAP reported in the literature, the ranges of XRD patterns usually are narrow to show strong reflections and do not show weak peaks at degrees above 60°, such as from 25 to 40° in Ref. 50 (Li, Y., et al. J. Mater. Sci.-Mater. Med. 1994, 5, 326–331.), from 25 to 35° in Ref. 39 (Ishikawa, K. et al. J. Mater. Sci.-Mater. Med. 1993, 4, 165–168.), from 10 to 50° in Ref. 49 (Zhou, J. et al. J. Mater. Sci.-Mater. Med. 1993, 4, 83–85.), from 10 to 60° in Ref. 40 (Bulina, N.V. et al. Minerals 2021, 11, 1310.), and so on.

    I added the following paragraph from Line 212, Page 6 in the revised manuscript.

    Figure S2 shows XRD patterns of HAP and Pd-PW-HAP before reaction. In the range from 5 to 70 degrees, the samples showed characteristic reflections of the HAP phase according to the Rigaku PDXL2 database (No 1011242). The strong reflections of HAP and Pd-PW-HAP existed in the range of 25‒45 degrees in the XRD patterns.

As for the peak at 40°of A, B, and D in the XRD patterns (Figure 5), I rewrote the following sentences in the manuscript.

Expression in the first manuscript: The reflection of Pd0 metal species could not be observed in the XRD pattern of Pd-PW-HAP after reaction.

Was Changed to:

 From Line 222, Page 7 in the revised manuscript

    Because a peak at 39.9 degrees corresponding to the (3 1 0) reflection of HAP appeared in the XRD pattern, the reflection of Pd (1 1 1) at 40.1 degrees could not be ensured in the XRD pattern of Pd-PW-HAP after reaction due to overlapping by the (3 1 0) reflection of HAP.

2. I added the TEM image of Pd-PW-HAP after the fifth circle reaction in the revised manuscript (Figure 6. TEM images of Pd-PW-HAP before reaction and after reaction. (A): Before reaction; (B): after reaction at 363 K for 8 h; (C): after reaction at 363 K for 8 h for 5 runs.).

I added the following sentences from Line 237, Page 7 in the revised manuscript:

    Moreover, the particle size of Pd metal did not increase after reaction at 363 K for 8 h for 5 runs.

    The EDS attached to the TEM instrument in this study does not have a mapping function, it only can analyze the elemental composition of the irradiated points in the TEM image.

     I added a new Table (Table S1. results of elemental analyses of Pd-PW-HAP before reaction by ICP and EDS) in the revised manuscript.

    I added the following sentences in the revised manuscript to explain the results of EDS.

    From Line 243, Page 7 in the revised manuscript:

    Table S1 shows the results of elemental analyses of Pd-PW-HAP before reaction by ICP and EDS. Three points on the Pd-PW-HAP particle in the TEM image of Pd-PW-HAP before reaction (Figure 6A) were analyzed using the EDS instrument attached to the electron microscope. The three points by EDS analyses gave a similar composition of each element, indicating that PW and Pd were uniformly distributed on the surface of Pd-PW-HAP before reaction. Because EDS mainly analyzed the solid surface of Pd-PW-HAP but ICP analyzed the HNO3 aqueous solution of Pd-PW-HAP, the EDS analyses gave lower values of Ca and P and higher values of W and Pd in comparison with the results of ICP analyses.

3.1). I added the following part from Line 333 on Page 10 in the revised manuscript.

    It has been reported that PdCl2 molecules can be fixed on the HAP surface and the resultant compounds are excellent catalysts for alcohol oxidation by O2 [44,47,48].

3.2). I added the following part from Line 366 on Page 11 in the revised manuscript.

    Not only the Pd2+ complex could be fixed on the surface of HAP [44], but also the Pd0 nanocluster formed from the reduction of Pd2+ during the reaction had a strong interaction with the surface of HAP [48]. This is the reason that Pd-PW-HAP could be used for five runs without a decrease in the catalytic activity (as shown in Table 3).

  1. Mori, K.; Hara, T.; Oshiba, M.; Mizugaki, T.; Ebitani, K; Kaneda, K. Catalytic investigations of carbon-carbon bond-forming reactions by a hydroxyapatite-bound palladium complex. New J. Chem. 2005, 29, 1174–1181.
  2. Mori, K.; Yamaguchi, K.; Hara, T.; Mizugaki, T.; Ebitani, K.; Kaneda, K. Controlled synthesis of hydroxyapatite-supported palladium complexes as highly efficient heterogeneous catalyst. J. Am. Chem. Soc. 2002, 124, 11572–11573.
  3. Hara, T.; Mori, K.; Oshiba, M.; Mizugaki, T.; Ebitani, K.; Kaneda, K. Highly efficient dehalogenation using hydroxyapatite-supported palladium nanocluster catalyst with molecular hydrogen. Green Chem. 2004, 6, 507–509.

   3.3). I added a new Figure (Figure 8. XPS spectra of Pd-PW-HAP before and after reaction) in the revised manuscript.

    I added the following part from Line 266 on Page 8 in the revised manuscript.

2.10. XPS spectra of Pd-PW-HAP before and after reaction

    Figure 8 shows the XPS spectra of Pd-PW-HAP before and after reaction. The binding energy of the Pd(3d5/2) peak was 338.2 eV in the XPS spectrum of Pd-PW-HAP before reaction, indicating that the Pd species remained in the oxidation state of +2 in the sample. On the other hand, the sample of Pd-PW-HAP after reaction showed the binding energy of the metallic Pd0 species with 3d5/2 of 334.9 eV and 3d3/2 of 340.2 eV in the XPS spectrum. These results proved that all Pd2+ species in the sample of Pd-PW-HAP before reaction were reduced into the metallic Pd0 species during the epoxidation of propylene by O2 in methanol at 363 K for 8 h.

    I added the following part to introduce the experiment of XPS

From Line 514 on Page 14 in the revised manuscript.

    X-ray photoelectron spectra (XPS) were carried out using a Phi-5500 ESCA spectrometer equipped with Mg Kα radiation (1253.6 eV). The chamber pressure was 109 Torr. The binding energy of Pd was calibrated by the C1s of 284.6 eV.

4. I added the EXAFS spectra in the revised manuscript (Figure 7. EXAFS functions of various samples. (a) Pd K-edge k3-weighted EXAFS spectra and (b) magnitude of Fourier transforms in the r-space.).

5. I added the GC charts in the revised manuscript (Figure S1. GC charts of products after reaction at 363 K for 8 h over Pd-PW-HAP.).

I added the following sentences from Line 121, Page 3 in the revised manuscript.

    Figure S1 shows the GC charts of gas and liquid products after reaction at 363 K for 8 h over Pd-PW-HAP.

6. I added the data of turnover numbers over various catalysts in Table 2 in the manuscript.

    I added the following sentences to explain turnover numbers in the manuscript.

    From Line 140, Page 4 in the revised manuscript.

    The turnover number (TON) of the catalyst was defined as the number of formed PO molecules per peroxy-oxygen bond [O–O]2– in the catalyst. The number of the peroxy-oxygen bond [O–O]2– was calculated using the catalyst’s W amounts, assuming one W molecule bonding with two [O–O]2– groups. The simultaneous existence of the peroxo-heteropoly compound and Pd is very important for improving the PO yield in the propylene epoxidation by O2 in methanol. As shown in Table 2, the PO yield and TON over the catalysts containing Pd and the peroxo-heteropoly compound decreased in an order of Pd+PW2 > Pd-PW-HAP > PdMgAl-PW4 > Pd-HMS-PW2 for the propylene epoxidation by O2 in methanol at 363 K. The homogeneous catalyst Pd+PW2 showed a higher PO yield and TON of [O–O]2– than the heterogeneous catalysts. On the other hand, Pd-PW-HAP obtained the highest PO yield and TON of [O–O]2– among various heterogeneous catalysts with the same loadings of Pd and W (1.2 wt.% of Pd and 13.2 wt.% of W).

Reviewer 4 Report

The authors presented the manuscript entitled “Immobilization of peroxo-heteropoly compound and palladium on hydroxyapatite for the epoxidation of propylene by molecular oxygen in methanol” peroxo-heteropoly hybrid compound was synthesized on the calcium-deficient 9 hydroxyapatite using a reaction of surface HPO4 groups with Na2[W2O3(O2)4]. The synthesized compound was characterized using several characterization techniques such as FT-IR. In general, the manuscript is well written. However, the manuscript is associated with minor issues and must be revised.  I advise the authors to take the following points into account when revising their manuscript.

Comment 1: Abstract should be more specific

Comment 2: The manuscript needs to be checked for typographical/ grammatical errors.

Comment 3: Include the Graphical Abstract in the revised manuscript.

Comment 4: In section 4.1, include the procured details and purity of all chemicals/materials used in the current study.

Comment 5: In FTIR, authors should fill in the peak wavenumber value of the band directly under the specific peak.

Comment 6: Table 3 values need to be converted into a graph to attain a broad readership. So draw the graph with the reusability conversions and selectivity obtained values.

Comment 7:  In the XRD analysis discussion (section 2.7.), include the peak positions and miller indices of the as-synthesized hybrid compound.

Comment 8: More recent references from the years 2022 - 2020 are required in the revised manuscript. 

Author Response

1: Abstract should be more specific

    I rewrote the Abstract in the revised manuscript. I added some experimental data in the Abstract to make it more specific.

2: The manuscript needs to be checked for typographical/ grammatical errors.

      I asked an English native speaker to check the English usage in the manuscript. I changed some English expressions in the revised manuscript.

3: Include the Graphical Abstract in the revised manuscript.

    I produced a Graphical Abstract in the revised manuscript.

4: In section 4.1, include the procured details and purity of all chemicals/materials used in the current study.

    I added the following part from Line 449, Page 13 in the revised manuscript.

4.1. Reagents

    Inorganic reagents were purchased from Wako Pure Chemical Industries Ltd. (Tokyo, Japan) with purities higher than 99%. Organic reagents were purchased from Tokyo Chemical Industry Co., Ltd. (Tokyo, Japan) with purities higher than 99.5%. Gas cylinders were purchased from Sumitomo Seika Chemicals Co., Ltd. (Tokyo, Japan) with purities higher than 99.995%.

5: In FTIR, authors should fill in the peak wavenumber value of the band directly under the specific peak.

    I filled in the peak wavenumber value in the FT-IR spectra (Figure 1. FT-IR spectra of various samples.) in the revised manuscript.

6: Table 3 values need to be converted into a graph to attain a broad readership. So draw the graph with the reusability conversions and selectivity obtained values.

    I added a new Figure (Figure 4. Time courses of solid catalyst Pd-PW-HAP in the propylene epoxidation by O2 in methanol for 5 runs.) in the revised manuscript.

    I added the following part to explain Figure 4 in the revised manuscript.

    From Line 192, Page 7 in the revised manuscript.

    Figure 4 shows the time courses of solid catalyst Pd-PW-HAP in the propylene epoxidation by O2 in methanol for 5 runs. The used catalyst was obtained through centrifugal separation and then dried under vacuum at room temperature for 3 h. When the amount of the used catalyst after drying was less than 0.5 g due to loss in the operation, a small amount of fresh catalyst was added to the used catalyst to keep the catalyst amount at 0.5 g for the next run. The propylene conversion greatly increased but the selectivity for PO slightly decreased with reaction time in each run. The catalytic activity of Pd-PW-HAP was almost kept in the reaction for 5 runs.

7:  In the XRD analysis discussion (section 2.7.), include the peak positions and miller indices of the as-synthesized hybrid compound.

    I added the miller indices in the XRD patterns (Figure 5) in the revised manuscript.

    Moreover, I added a new Figure with degrees from 5 to 70 in the XRD patterns (Figure S2. XRD patterns of HAP and Pd-PW-HAP before reaction) and indicated the miller indices of main reflections according to the reference database in the revised manuscript.

I discussed the XRD patterns using peak positions and miller indices as follows.

From Line 213, Page 6 in the revised manuscript:

    Figure S2 shows XRD patterns of HAP and Pd-PW-HAP before reaction. In the range from 5 to 70 degrees, the samples showed characteristic reflections of the HAP phase according to the Rigaku PDXL2 database (No 1011242). The strong reflections of HAP and Pd-PW-HAP existed in the range of 25‒45 degrees in the XRD patterns.

    Figure 5 shows the XRD patterns of Pd-PW-HAP and Pd+PW2 before and after reaction. The sample of Pd+PW2 before reaction showed the reflections of the peroxo-heteropoly compound in the XRD pattern. For the sample of Pd+PW2 after reaction at 363 K for 8 h, a weak peak at 40.1 degrees corresponding to the reflection of (1 1 1) for Pd0 metal species could be observed in the XRD pattern. The XRD pattern of Pd-PW-HAP after reaction at 363 K for 8 h was almost the same as that of Pd-PW-HAP before reaction. Because a peak at 39.9 degrees corresponding to the (3 1 0) reflection of HAP appeared in the XRD pattern, the reflection of Pd (1 1 1) at 40.1 degrees could not be ensured in the XRD pattern of Pd-PW-HAP after reaction due to overlapping by the (3 1 0) reflection of HAP.

8: More recent references from the years 2022 - 2020 are required in the revised manuscript. 

    I investigated the newest literature and added the following references in the revised manuscript.

27. Kruse, J.H.; Langer, M.; Romanenko, I.; Trentin, I.; Hernández-Castillo, D.; González, L.; Schacher, F.H.; Streb, C. Polyoxometalate-soft matter composite materials: Design strategies, applications, and future directions. Adv. Funct. Mater. 2022, 2208428.

28. Buru C.T.; Farha O.K. Strategies for incorporating catalytically active polyoxometalates in metal−organic frameworks for organic transformations. ACS Appl. Mater. Interfaces 2020, 12, 5345−5360.

30. Gao, Y.; Mirante, F.; de Castro, B.; Zhao, J.; Cunha-Silva, L.; Balula, S.S. An effective hybrid heterogeneous catalyst to desulfurize diesel: Peroxotungstate@Metal–organic framework. Molecules2020, 25, 5494.

31. Chilivery, R.; Chaitanya, V.; Nayak, J.; Seth, S.; Rana, R.K. Heterogenization of phosphotungstate clusters into magnetic microspheres: Catalyst for selective oxidation of alcohol in water. ACS Sustainable Chem. Eng. 2022, 10, 6925−6933.

40. Bulina, N.V.; Makarova, S.V.; Baev, S.G.; Matvienko, A.A.; Gerasimov, K.B.; Logutenko, O.A.; Bystrov, V.S. A Study of Thermal Stability of Hydroxyapatite. Minerals 2021, 11, 1310.

55. Bulina, N.V.; Eremina, N.V.; Vinokurova, O.B.; Ishchenko, A.V.; Chaikina, M.V. Diffusion of copper ions in the lattice of substituted hydroxyapatite during heat treatment. Materials 2022, 15, 5759.

56. Bulina, N.V.; Rybin, D.K.; Makarova, S.V.; Dudina, D.V.; Batraev, I.S.; Utkin, A.V.; Prosanov, I.Y.; Khvostov, M.V.; Ulianitsky, V.Y. Detonation Spraying of Hydroxyapatite on a Titanium Alloy Implant. Materials 2021, 14, 4852.